# Premature mortality and years of potential life lost from cardiovascular diseases: Protocol of a systematic review and meta-analysis

**Wan Shakira Rodzlan Hasani**[1,2☯*], **Nor Asiah Muhamad**[3☯], **Nur Hasnah Maamor**[3],
**Tengku Muhammad Hanis**[1], **Chen Xin Wee**[4], **Muhammad Radzi Abu Hassan**[5],
**Zulkarnain Abdul Karim**[6], **Kamarul Imran Musa**[1]

**1** Department of Community Medicine, School of Medical Sciences, Universiti Sains Malaysia, Kubang Kerian, Kelantan, Malaysia, **2** Institute for Public Health, National Institutes of Health, Ministry of Health Malaysia, Setia Alam, Selangor, Malaysia, **3** Sector for Evidence-based Healthcare, National Institutes of Health, Ministry of Health, Shah Alam, Selangor, Malaysia, **4** Faculty of Medicine, Department of Public Health Medicine, Universiti Teknologi MARA, Sungai Buloh Campus, Selangor, Malaysia, **5** Office of Deputy Director-General, Ministry of Health Malaysia, Putrajaya, Malaysia, **6** Office of The Manager, National Institutes of Health, Ministry of Health Malaysia, Setia Alam, Selangor, Malaysia

☯ These authors contributed equally to this work.
* shaki_iera@yahoo.com

**Data Availability Statement:** No datasets were generated or analysed during the current study. All

## Abstract

### Introduction

Despite the burden of cardiovascular disease (CVD) continuing to increase globally, no comprehensive meta-analyses have been conducted quantifying premature CVD mortality. This paper reports the protocol for a systematic review and meta-analysis to derive updated estimates of premature CVD mortality.

### Methods and expected outputs

This review will include the studies that reported premature CVD mortality based on standard premature mortality indicators, including years of life lost (YLL), age standardized mortality rate (ASMR) or standardised mortality ratio (SMR). PUBMED, Scopus, Web of Science (WoS), CINAHL, and Cochrane Central Register of Controlled Trials (CENTRAL) will be used as the literature databases. The study selection as well as the evaluation of the quality of the included articles will be done independently by two reviewers. Pooled estimates of YLL, ASMR, and SMR will be computed by applying random-effects meta-analysis. Heterogeneity among selected studies will be assessed using the $I^2$ statistic and Q statistic with associated p-values. A funnel plot analysis and Egger's test will be conducted to assess the potential impact of publication bias. Depending on data availability, we propose to conduct subgroup analyses by sex, geographic location, main CVD types, and study time. Reporting of our findings will follow the Preferred Reporting Items for Systematic Review and Meta-Analyses (PRISMA) guidelines.

relevant data from this study will be made available upon study completion.

**Funding:** The author(s) received no specific funding for this work.

**Competing interests:** The authors have declared that no competing interests exist.

## Conclusion

Our meta-analysis will provide a comprehensive synthesis of the available evidence on premature CVD mortality, which is a major public health concern worldwide. The results of this meta-analysis will have important implications for clinical practice and public health policy, providing insights into strategies to prevent and manage premature CVD mortality.

## Trial registration

**Systematic review registration:** PROSPERO CRD42021288415. https://www.crd.york.ac.uk/prospero/display_record.php?ID=CRD42021288415.

## Introduction

Premature mortality is defined as a death that occurs before the average age of death in a particular population [1]. It is a measure of unfulfilled life expectancy. According to the Global Burden of Disease (GBD) study, cardiovascular diseases (CVDs), principally ischemic heart disease (IHD) and stroke, remain the leading causes of premature mortality worldwide [2]. Together with cancer, respiratory disease, and diabetes, they represent more than 80% of all premature mortality due to non-communicable diseases (NCD) [3]. The World Health Organization (WHO) reported the burden of premature mortality was notably high in low- or middle-income countries (LMICs) [4]. Although CVD mortality rates have seen dramatic declines in the past two decades, LMICs are confronted by an increasing number of people experiencing premature CVD mortality [5].

NCDs, especially CVD, are currently at epidemic levels. The need to track their impacts on premature mortality is a key goal of health care and public health programs. Using GBD study data from 1990 and 2013, Gregory et al. projected about 7.8 million premature CVD deaths in 2025, if the current risk factors trends (including hypertension, tobacco smoking, diabetes mellitus, and obesity) continue [2]. In 2015 the WHO developed an ambitious target by 2030 to reduce by one-third premature mortality from NCDs through the Sustainable Development Goal (SDG) [6]. The existing policies for the prevention and control of NCDs, including CVDs, must be revised urgently [7]. In order to ensure the SDG's target is on track, information on premature CVD mortality may assist in the development of global and context-specific strategies for reducing the incidence of premature CVD mortality.

There are several methods to calculate the burden of premature mortality. A year of potential life lost (YPLL) is a standardised parameter commonly used to measure the burden of disease due to premature mortality [8,9]. YPLL was calculated by multiplying the number of deaths at each age by the number of potential years remaining for that age. Therefore, in this situation, a mostly arbitrary cut-off age must be chosen [10], where a decision might be made based on the population's average present life expectancy. For example, the Organisation for Economic Co-operation and Development (OECD) calculates premature mortality as YPLL before the age of 70 [11]. The GBD study calculated years of life lost by multiplying the number of deaths by a global standard life expectancy at the age of death and employing a standard term, "standard expected years of life lost" (SEYLL) [12]. On the other hand, age-standardized mortality rate (ASMR) [13], is a measure that allows for the comparison of mortality rates between populations with different age structures. ASMR adjusts for differences in age distribution by applying age-specific death rates from a standard population to the age distribution of the population of interest. In the context of premature mortality, ASMR can provide a more

accurate picture of the burden of disease across populations with different age structures and can help identify populations with the highest rates of premature CVD mortality. The WHO considers an ASMR of 30 to 70 years for premature mortality [12], whereas some studies favour ASMRs of 65 to 75 years [14,15]. The standard mortality ratio (SMR) is also frequently used to compare the mortality risk, including premature mortality, of a study population to that of a standard or reference population [16]. This method establishes a fixed age below which any death is considered "premature."

Several systematic reviews and meta-analyses have been conducted to investigate CVD mortality but have not specified premature mortality as an outcome measure. These analyses have typically focused on establishing the predictors of increased mortality, relative risks, or risk factors for cause-specific mortality [17–20]. Meanwhile, the GBD study, which is the most widely used global estimate of premature mortality, derived the estimate of global premature CVD mortality with missing or limited quality mortality data in some countries, particularly in the poorest regions [21,22]. To the best of our knowledge, there has not yet been a comprehensive systematic review and meta-analysis that has synthesized the available evidence on premature CVD mortality to estimate a pooled effect size. Thus, we planned to fill these gaps by conducting a systematic review and meta-analysis in which we aimed to identify studies and synthesise their findings on YPLL, SEYLL, ASMR, and SMR for premature CVD mortality. If applicable, we aimed to stratify the findings by sex, geographical location, main CVD types, and study time. This protocol is designed to establish and synthesise information on premature CVD mortality. We will explicitly describe the methodology for conducting a systematic review and meta-analysis for this review.

## Review questions

1. What are the pooled estimates of premature CVD mortality using YPLL, SEYLL, ASMR, and SMRs?

2. What are the pooled estimates of premature CVD mortality according to sex, geographical areas (six continents including Asia, Africa, Europe, North America, South America, and Oceania), the two main CVD types (ischemic heart disease and cerebrovascular disease), and timing of the study?

## Method

This protocol was registered with the International Prospective Register of Systematic Reviews (PROSPERO), [registration number: CRD42021288415]. This systematic review will be reported and conducted following the Preferred Reporting Items for Systematic Reviews and Meta-analysis (PRISMA) protocol [23] (S1 Checklist). Simultaneously, we will integrate this report with the Meta-analysis of Observational Studies in Epidemiology (MOOSE) guideline [24] since the outcome of this review will be a meta-analysis of selected observational studies.

### Eligibility criteria

We will include all original articles in English as follows;

1. Report data on years of life lost (either using the YPLL or SEYLL method), ASMR or SMR as indicators for premature CVD mortality measurement,

2. Measure premature CVD mortality with any upper limit on age or age range to define premature mortality.

3. The cause of CVD death should be determined using the International Classification of Diseases (ICD) code (any version).

4. Investigate modifiable risk factors associated with premature CVD mortality, including cardio-metabolic or behavioural risk factors such as diabetes, hypertension, hypercholesterolemia, obesity, tobacco smoking, alcohol use, an unhealthy diet, physical inactivity, or low socioeconomic status (low income, low education level, and employment status), will be considered.

5. Any observational study design (including cross-sectional, cohort, and case control studies) or intervention studies that report premature CVD mortality. We will combine all observational studies that provided YLL, ASMR, or SMR measurements and pool the results. We will also combine the intervention studies that measure the premature mortality indicators and pool their findings. The selected studies will be evaluated using quality assessment tools based on appropriate study design before being included in the analysis. The results of the observational and interventional studies will be presented in separate tables.

The exclusion criteria are;

1. Studies that assess causes of premature mortality other than cardiovascular disease.

2. Any review, case study, commentary, or qualitative studies.

3. Very specific population (e.g., among some disease conditions such as epilepsy, congenital disease, the post-surgical group, pregnant women, etc.).

## Outcome measures

The following are the outcomes of this review:

1. The pooled estimates of YLL due to premature CVD mortality. The method of calculating YLL will be collected from selected articles. We will include the original method of Gardner (1990) [10] for YPLL calculation and the SEYLL method from the GBD study [25]. The YPLL method calculates the number of deaths at each age by multiplying the number of deaths by an indicator of years of potential life remaining for that age, and the terms are summed to get the total number years of potential life lost. The SEYLL formula is based on comparing the age of death to the standard life expectancy of an individual at each age. The details of the formulas for YPLL and SEYLL are presented below;

$$YPLL = \sum_{i=0}^{N} di(N - i)$$

Where, $i$ is age at death, $di$ is number of deaths at age $i$, and
N is upper cut-off age.

$$SEYLL = \sum_{x=0}^{l} d_x e_x^*$$

Where, $d_x$ is the number of deaths and $e_x^*$ is expected years of life at each age in the standard population

2. The pooled estimates of ASMR from premature CVD mortality. It is a measure of the mortality rate in a population that has been adjusted for differences in the age distribution of the population. ASMR calculation entailed two steps [26]: 1) calculate age-specific mortality

rates. This involves calculating the number of deaths in each age group and dividing it by the corresponding population size for that age group. The result is the age-specific mortality rate. 2) Apply the standard population structure. The age-specific mortality rates are then multiplied by the corresponding proportion of the standard population in each age group. The ASMR is then calculated as the ratio of the expected number of deaths to the corresponding standard population size, expressed as a rate per 100,000 or 1,000 population. The formula for calculating ASMR is as follows:

$$ASMR = \frac{(\Sigma \ (\text{age} - \text{specific mortality rate} \times \text{standard population proportion})}{(\text{Standard population size})} \times 100,000$$

where $\Sigma$ represents the sum over all age groups.

3. The pooled estimate of SMR for premature CVD mortality. The SMR gives the ratio of deaths that are due to CVD compared to the general population. For each cause of death, SMRs and their 95% confidence intervals will be extracted from each publication. If not reported, SMR will be calculated using this formula [16]:

$$SMR = \frac{O \ (\text{Observed number of deaths})}{E \ (\text{Expected number of deaths})}$$

## Information sources

**Electronic search.** We will systematically conduct a comprehensive literature search using various literature databases, including PubMed, Scopus, Web of Science (WoS), the Cumulative Index to Nursing and Allied Health Literature (CINAHL), and Cochrane Central Register of Controlled Trials (CENTRAL), to identify eligible studies. Secondary searches will be conducted in other sources, such as Google Scholar. The reference section of the included studies will also be hand-searched for additional relevant studies. Studies will be restricted to the English language, and there is no restriction on the publication date. The search will be performed from selected electronic databases up to August 2022.

**Search strategy.** The proposed search term for the first theme will be "cardiovascular diseases", including keywords for coronary heart disease, cerebrovascular disorder, myocardial ischemia, and stroke. The second theme is "modifiable risk" or "behavioural risk", including hypertension, hyperglycemia, hyperlipidaemia, obesity or overweight, tobacco use, physical inactivity, an unhealthy diet, alcohol use, and low socioeconomic status. The third theme is "premature mortality", including premature death, years of life lost, potential years of life lost, age standardize mortality rate and "standardize mortality ratio". The exploded versions of Medical Subject Headings (MeSH) for each theme will be included. All the three search themes will be combined using the Boolean operator "AND". The detailed search terms for each database are presented in S1 Table.

**Study selection.** Two review authors (WSRH and HM) will independently screen all the titles and abstracts to examine the potential studies for inclusion and exclude those that are obviously irrelevant. We will identify the studies and code them as "retrieve" (eligible or potentially eligible/unclear) or "do not retrieve". We will retrieve the full-text study reports and publications, and the review authors (WSRH and CXW) will independently screen the full text to identify studies for inclusion, as well as identify and record reasons for the exclusion of the ineligible studies. We will resolve any disagreement through discussion and try to reach a decision. If no consensus could be reached, the two remaining authors (NAM and KIM) could act as arbiters.

We will identify and exclude duplicates and collate multiple reports of the same study so that each study, rather than each report, is the unit of interest in the review. We will record the selection process in sufficient detail to complete a PRISMA flow diagram and construct a table describing the characteristics of the excluded studies [23,27]. The Mendeley Reference Management Software [28] will be used to store, organize, and manage all the references.

*Data extraction and management.* We will use a standardized data extraction form created by the Microsoft Excel Spreadsheet Software for study characteristics and outcome data. Two review authors (WSRH, TMH) will independently extract outcome data from the included study. We will note in the "characteristics of included studies" table if outcome data is not reported in a usable way. We will resolve any disagreements by consensus or by involving the two remaining authors (NAM and KIM). We will double-check that data is entered correctly by comparing the data present in the systematic review with the study reports. The following study characteristics from the included studies will be extracted;

- Title, authors, study county, region, and publication year.

- Methods: study design, source of data, total duration of the study, and method of analysis (including formula or definition used for YLL, ASMR, and SMRs).

- Participants: total number of deaths (n), age range, and sex.

- Outcome: total number of YLL, the YLL rate, the ASMR rate, and the SMR for premature CVD mortality with a measure of uncertainty (e.g., confidence interval or standard error). We will contact the authors to obtain premature CVD mortality estimates if premature mortality was reported as part of all-cause or general NCD mortality.

- Exposures or risks: whenever data is applicable, we will extract the data on exposure risks such as types of CVD, comorbidities, or any modifiable risk factors, control conditions, and adjustment variables.

*Quality assessment.* Two review authors (WSRH and NAM) will independently assess the study's quality based on the criteria in the Newcastle-Ottawa Scale (NOS) for observational studies. The NOS is a widely used tool for assessing the quality of non-randomized studies, including cohort studies and case-control studies, in systematic reviews and meta-analyses. [29]. We will include studies with a potentially high risk of bias. We will assign each potential source of bias as high, low, or unclear and provide a quote from the study report together with justification for our assignment in the quality assessment table included studies. NOS applied a "star system," where the study is assessed based on three broad perspectives: 1) the selection of the study groups; 2) the comparability of the groups, and 3) the ascertainment of exposure and outcome [29]. The maximum score is 9 points, where the studies can be classified as good, fair, or poor quality according to the following standard thresholds; -

1. Good quality: 3 or 4 stars in the selection domain AND 1 or 2 stars in the comparability domain AND 2 or 3 stars in outcome/exposure domain.

2. Fair quality: 2 stars in the selection domain AND 1 or 2 stars in the comparability domain AND 2 or 3 stars in outcome/exposure domain.

3. Poor quality: 0 or 1 star in the selection domain OR 0 stars in the comparability domain OR 0 or 1 star in the outcome/exposure domain.

To assess the risk of bias in the included intervention studies, we will use the Cochrane Risk of Bias Assessment Tool: for Non-Randomized Studies of Interventions (ACROBAT- NRSi) tool [30]. This assessment tool consists of five domains of bias, including selection bias,

confounding bias, performance bias, detection bias, and attrition bias, which will be assessed using a set of signalling questions. Each domain will be rated as having a low, high, or unclear risk of bias, and an overall risk of bias rating will be assigned for each study. The use of the ACROBAT- NSRi tool will allow us to provide a standardized assessment of the quality of evidence from non-randomized studies of interventions and to determine the overall risk of bias across studies.

## Statistical analysis

**Data analysis and statistical analysis.** Statistical analysis will be conducted using R software. The main R packages, "meta" [31] and "metafor" [32], will be used for meta-analysis. We will calculate the pooled estimates of premature mortality for YPLL, SEYLL, ASMR, and SMR using a random effect model to allow for heterogeneity across studies. Whenever data is applicable, we also plan to perform stratification of estimates by sex, the two major CVD types (ischemic heart disease and cerebrovascular disease), geographical area, and study time.

For YPLL and SEYLL, the weighted average with 95% CI (if reported in the original articles) will be calculated. If the uncertainty (SEs or CIs) are not reported, the usual variance-based method could not be used to calculate study weights. Thus, we will calculate the average values weighted based on the size of the individual study population. If we find studies that report YPLL or SEYLL estimates by subgroups, we will calculate the overall mean. Whenever numbers of deaths by subgroup are available, we will calculate the weighted estimates of YPLL or SEYLL based on the number of patients who die in each group.

Since the ASMRs or SMRs are often corrected for age and sex, we will use the most basic adjusted model when several estimates are reported. If the included studies provide numerous estimates per subgroup, we will combine the rates or ratios according to the procedures outlined by van Dooren et al. [33] to obtain a single estimate for each study and analysis.

The results will be separated and evaluated on the basis of the study design and methods used for premature mortality calculation, which are YPLL, SEYLL, ASMR, and SMR. For each pool estimate, a forest plot will be drawn, and the distribution will be presented graphically.

*Assessment of heterogeneity.* Assessing heterogeneity is a critical step in conducting a meta-analysis, as it allows for an evaluation of the degree of inconsistency or variability among the results of individual studies. For this review, we will evaluate heterogeneity using both the $I^2$ and Q statistics. The $I^2$ statistic will be used to quantify the impact of heterogeneity, with percentage of around 25%, 50%, and 75% representing a low, moderate, and high degree of heterogeneity, respectively [34]. The Q-test is a statistical test that determines whether there is significant heterogeneity among the studies. The significance level for the Q test is set at 0.01 [35] in this review. If significant heterogeneity is detected using the Q-test or $I^2$ index ($> 50\%$), we will explore potential sources of heterogeneity using subgroup analyses and meta-regression. We will also explore the possible causes (for example, differences in study quality, participants, or outcome assessments) and evaluate the studies in terms of their methodological characteristics to determine whether the degree of heterogeneity can be explained by differences in those characteristics and whether a meta-analysis is appropriate.

*Assessment of publication bias.* We will create and examine a funnel plot to explore possible small study biases if we can pool more than ten studies in a single meta-analysis. The number of studies that are missing from the funnel plot will be estimated. The effect size after the imputation of these missing studies will be estimated by the trim-and-fill method. The trim and fill method is a simple estimation approach proposed by Duval and Tweedie [36], where they trim off the asymmetric outlying part of the funnel, then use the symmetric remainder to estimate the true center of the funnel, and then replace the trimmed studies and their counterparts

around the centre. Other methods to assess the publication bias, including Begg's rank correlation [37] and Egger's weighted regression method test [38] will also be performed.

## Ethics and dissemination

We registered this systematic review with the National Medical Research Register (NMRR), Ministry of Health Malaysia (NMRR ID-22-00231-MOX). Since this review will use published data, ethical approval is waived. The systematic review will focus on premature CVD mortality and its associated factors, the results of which will be disseminated by publication in a peer-reviewed journal after completion.

## Discussion

The increasing burden of premature death due to CVD warrant the researchers to obtain the updated analysis on the global and setting-specific (e.g., difference geographical or subgroup population) of premature CVD mortality. Despite the growing number of individual studies that reported YLL, ASMR or SMR for premature mortality [39–42], the authors' limited search revealed no recent meta-analysis focusing on CVD-related premature death.

To achieve the ambitious SDG plan, target 3.4 (by 2030, to reduce by one third premature mortality from NCD), data on premature CVD mortality and its predictors will have a role in tracking the progress towards achieving the target. We strongly believe that our conclusion will be of crucial importance for both research and future policy-making, especially in helping to formulate more effective and targeted NCD prevention strategies. Our findings could also help in the development of mathematical models or cost-effectiveness analyses to forecast the future premature death burden from CVD. The data might help to identify settings or subgroups of the population where the risk of CVD death is of higher concern and needs special prevention priorities (e.g., on a specific continent or gender-based interventions). Future studies could be conducted to explore the modifiable determinants of CVD in certain high-risk subgroups or settings.

Strengths and limitations will be highlighted in the process of identifying evidence. This systematic review will yield solid and updated estimates of the global prevalence of premature CVD mortality. Unlike previous global reports, the data included will not be limited to studies conducted in high-income countries, and records from LMICs will be included. The meticulous design, use of standardised study rating instruments, and compliance with all relevant guidelines for systematic reviews and meta-analyses are the anticipated strengths. Limitations will mainly originate from the different designs and characteristics of all the included studies; this may lead to high heterogeneity, which, in turn, will lower the quality of the evidence of this meta-analysis and systematic review. However, this may be overcome by including subgroup analyses and meta-regression in the meta-analysis. Our review will also include studies based on observational designs. Observational studies can be more prone to bias and confounding. Synthesizing evidence from multiple observational studies, however, can strengthen the conclusions that can be drawn.

## Supporting information

**S1 Checklist. PRISMA checklist.**
(DOC)

**S1 Table. Proposed search terms.**
(DOCX)

## Acknowledgments

We would like to thank the Director-General of Health Malaysia for his permission to publish this article.

## Author Contributions

**Conceptualization:** Wan Shakira Rodzlan Hasani, Nor Asiah Muhamad, Tengku Muhammad Hanis, Muhammad Radzi Abu Hassan, Zulkarnain Abdul Karim, Kamarul Imran Musa.

**Investigation:** Wan Shakira Rodzlan Hasani, Nur Hasnah Maamor, Tengku Muhammad Hanis, Chen Xin Wee.

**Methodology:** Wan Shakira Rodzlan Hasani, Nor Asiah Muhamad, Tengku Muhammad Hanis, Kamarul Imran Musa.

**Project administration:** Wan Shakira Rodzlan Hasani, Nor Asiah Muhamad, Nur Hasnah Maamor.

**Resources:** Wan Shakira Rodzlan Hasani, Nor Asiah Muhamad.

**Supervision:** Nor Asiah Muhamad, Tengku Muhammad Hanis, Kamarul Imran Musa.

**Validation:** Nor Asiah Muhamad, Tengku Muhammad Hanis, Kamarul Imran Musa.

**Visualization:** Wan Shakira Rodzlan Hasani, Tengku Muhammad Hanis.

**Writing – original draft:** Wan Shakira Rodzlan Hasani.

**Writing – review & editing:** Wan Shakira Rodzlan Hasani, Nor Asiah Muhamad, Nur Hasnah Maamor, Tengku Muhammad Hanis, Chen Xin Wee, Muhammad Radzi Abu Hassan, Zulkarnain Abdul Karim, Kamarul Imran Musa.

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
