## [Decision Letter · Decision Letter 0]

15 Feb 2023

PONE-D-22-21753Premature mortality and years of potential life lost from cardiovascular diseases: protocol of a systematic review and meta-analysisPLOS ONE

Dear Dr. Rodzlan Hasani,

Thank you for submitting your manuscript to PLOS ONE. After careful consideration, we feel that it has merit but does not fully meet PLOS ONE’s publication criteria as it currently stands. Therefore, we invite you to submit a revised version of the manuscript that addresses the points raised during the review process. ***If there are any similar comments, please address comments of both reviewers. You may mention that the edit was already done for reviewer ''n'', comment ''x''. ***

We look forward to receiving your revised manuscript.

Kind regards,

Ahmed Mustafa Rashid

Academic Editor

PLOS ONE

Journal Requirements:

2. Please include a caption for figure 1.

Reviewers' comments:

Reviewer's Responses to Questions

**Comments to the Author**

1. Does the manuscript provide a valid rationale for the proposed study, with clearly identified and justified research questions?

Reviewer #1: Yes

Reviewer #2: Yes

Reviewer #3: Yes

2. Is the protocol technically sound and planned in a manner that will lead to a meaningful outcome and allow testing the stated hypotheses?

Reviewer #1: Partly

Reviewer #2: Yes

Reviewer #3: Yes

3. Is the methodology feasible and described in sufficient detail to allow the work to be replicable?

Reviewer #1: Yes

Reviewer #2: Yes

Reviewer #3: Yes

4. Have the authors described where all data underlying the findings will be made available when the study is complete?

Reviewer #1: Yes

Reviewer #2: Yes

Reviewer #3: Yes

5. Is the manuscript presented in an intelligible fashion and written in standard English?

Reviewer #1: Yes

Reviewer #2: No

Reviewer #3: Yes

6. Review Comments to the Author

You may also provide optional suggestions and comments to authors that they might find helpful in planning their study.

Reviewer #1: The protocol identifies a relevant gap in the existing evidence.

Authors have proposed to combine observational studies but the adjustments for combining cohort, case control and cross-sectional studies are not mentioned in detail. As these designs address different questions it is important to know how authors plan to address the differences due to study design.

Its not unusual to combine observational with intervention data but there are very few details about bringing the evidence from these designs together in the protocol. Details about the analysis plan should be included.

There's little or no information about assessing the risk of bias in the interventional studies.

Overall its an interesting protocol and I wish authors all the best for their review.

Reviewer #2: Wan et al. described a study protocol on “Premature mortality and years of potential life lost from cardiovascular diseases: protocol of a systematic review and meta-analysis” in order to identify studies and synthesize their findings on years of potential life lost (YPLL) and standard mortality ratios (SMRs) for premature cardiovascular diseases mortality. However, in my opinion, the manuscript can be improved by incorporating the following edits:

1) In the abstract, authors are requested to precisely summarize each section including the introduction, objectives, methods, and discussion for better clarity for the readers.

2) In the introduction, the authors have described, ‘’Although CVD mortality rates have seen dramatic declines in the past two decades (1990 - 2019)’’. They are requested to please rephrase the statement by either removing the phrase 'past two decades' or '1990-2019' since they do not complement each other.

3) In the introduction, the authors stated, ‘’A year of potential life lost (YPLL) is a standardised parameter commonly used to measure the burden of disease due to premature mortality.’’ This information should be evidenced with a reference for the authenticity of the content.

4) In the introduction, the authors have written ‘OECD.’ It is advised to please write the expanded form, as it is used in the manuscript for the first time.

5) In the search strategy, the authors have described ‘’All the four search themes will be combined.’’ However, only 3 themes have been defined above.

6) In the study selection, authors could consider removing the link to Mendeley reference management software as a reference has already been provided.

7) The authors are advised to use only abbreviations throughout the manuscript if the expanded forms have already been described. For example, writing ‘NCDs’ instead of ‘non-communicable diseases’ again.

8) The authors are advised to please proofread their manuscript as it contains a few grammatical and punctuation errors and some inappropriately structured sentences.

Reviewer #3: Hasani et al have formulated a protocol of a systematic review and meta-analysis on “Premature mortality and years of potential life lost from cardiovascular diseases” to derive updated estimates of years of potential life lost (YPLL) due CVD and standardized mortality ratios (SMRs) of premature CVD mortality. In my opinion, a few edits can be incorporated to further improve the manuscript:

1. The ‘abstract’ and ‘statistical analysis’ sections have no mention of ‘p values’. The authors should provide information of what p value will be considered as significant to highlight the significance of the results this study will produce.

2. In point 1 of the inclusion criteria, please use the abbreviation of ‘years of potential life lost’ as YPLL since the expanded form has been used earlier in the manuscript.

3. From point 4 of the inclusion criteria, the authors should consider removing “We will also consider including any” from the rest of the phrase for maintaining the structure of this section.

4. In the ‘discussion’ section, the authors have described the limitations of using observational studies as “Such studies have limitations in drawing precise inferences”. However, this is a vague point and can be replaced by “observational studies can be more prone to bias and confounding”.

7. PLOS authors have the option to publish the peer review history of their article (what does this mean?). If published, this will include your full peer review and any attached files.

Reviewer #1: No

Reviewer #2: **Yes: **Mohammad Arham Siddiq

Reviewer #3: No

---

## [Author Response · Author response to Decision Letter 0]

28 Feb 2023

Response to reviewer’s 1 comments

The protocol identifies a relevant gap in the existing evidence.

Author’s response

Dear reviewer, we are grateful for the comment. In response to this comment, we have revamped this review as follows:

1) Authors have proposed to combine observational studies but the adjustments for combining cohort, case control and cross-sectional studies are not mentioned in detail. As these designs address different questions it is important to know how authors plan to address the differences due to study design.

Thank you for comment. We agree with your point. Thus, we added a few statements about combining all observational studies under the inclusion criteria (no. 5).

2) Its not unusual to combine observational with intervention data but there are very few details about bringing the evidence from these designs together in the protocol. Details about the analysis plan should be included.

We appreciate your comment. We will present separately the outcomes from the observational study and the interventional study. We added this statement under inclusion criteria (no. 5). We also added the plan for analysis under "Data Analysis and Statistical Analysis" (last paragraph).

3) There's little or no information about assessing the risk of bias in the interventional studies.

Thank you very much for your comments. We added the measurement tools (ACROBAT-NRSi) under section

Quality assessment (last paragraph). 

4) Overall its an interesting protocol and I wish authors all the best for their review.

Thank you for your positive feedback on our protocol. We appreciate your encouragement and support for our research. We are committed to conducting a high-quality review and providing valuable insights into the topic of interest. We will do our best to address any concerns or suggestions you may have to further improve the protocol and ensure a rigorous and comprehensive review. Thank you for your time and consideration.

Response to reviewer’s 2 comments

Wan et al. described a study protocol on “Premature mortality and years of potential life lost from cardiovascular diseases: protocol of a systematic review and meta-analysis” in order to identify studies and synthesize their findings on years of potential life lost (YPLL) and standard mortality ratios (SMRs) for premature cardiovascular diseases mortality. However, in my opinion, the manuscript can be improved by incorporating the following edits:

Author’s response

Dear reviewer, we greatly appreciate your feedback. We have provided our point-by-point response to each of your comments below.

1) In the abstract, authors are requested to precisely summarize each section including the introduction, objectives, methods, and discussion for better clarity for the readers.

We appreciate your comment. We revised our abstract as suggested. We hope that the revised abstract has met your expectations.

2) In the introduction, the authors have described, ‘’Although CVD mortality rates have seen dramatic declines in the past two decades (1990 - 2019)’’. They are requested to please rephrase the statement by either removing the phrase 'past two decades' or '1990-2019' since they do not complement each other.

Thank you for your comment. We changed the sentences by removing “1990-2019” based on your suggestions. 

3) In the introduction, the authors stated, ‘’A year of potential life lost (YPLL) is a standardised parameter commonly used to measure the burden of disease due to premature mortality.’’ This information should be evidenced with a reference for the authenticity of the content. 

Thank you for your suggestion. We added citation for this statement.

4) In the introduction, the authors have written ‘OECD.’ It is advised to please write the expanded form, as it is used in the manuscript for the first time. 

Appreciate your suggestion. We added the abbreviation for OECD (Organisation for Economic Co-operation and Development) in text. 

5) In the search strategy, the authors have described ‘’All the four search themes will be combined.’’ However, only 3 themes have been defined above. 

We appreciated your suggestion. Apologies for overlooking this. We changed the sentence into “All the three search….”

6) In the study selection, authors could consider removing the link to Mendeley reference management software as a reference has already been provided. 

Thank you for your comment. The link for Mendeley reference was removed. 

7) The authors are advised to use only abbreviations throughout the manuscript if the expanded forms have already been described. For example, writing ‘NCDs’ instead of ‘non-communicable diseases’ again. 

Thank you for the suggestion. We agreed with your point. We changed ‘non-communicable diseases’ to NCD throughout the manuscript. 

8) The authors are advised to please proofread their manuscript as it contains a few grammatical and punctuation errors and some inappropriately structured sentences. 

Thank you for taking the time to review our manuscript. 

We apologize for any grammatical and punctuation errors that were present in the initial submission. We have conducted a thorough proofreading of the manuscript and made the necessary corrections. Additionally, we have restructured some of the sentences to improve their clarity and flow. We hope that these changes have addressed the concerns you raised regarding the appropriateness of the sentence structure.

Response to reviewer’s 3 comments

Hasani et al have formulated a protocol of a systematic review and meta-analysis on “Premature mortality and years of potential life lost from cardiovascular diseases” to derive updated estimates of years of potential life lost (YPLL) due CVD and standardized mortality ratios (SMRs) of premature CVD mortality. In my opinion, a few edits can be incorporated to further improve the manuscript:

Author’s response

Dear reviewer, we are very grateful for your comments. Herewith we represent our point-by-point response to each comment made by you.

1) The ‘abstract’ and ‘statistical analysis’ sections have no mention of ‘p values’. The authors should provide information of what p value will be considered as significant to highlight the significance of the results this study will produce. 

Thank you very much for your comment. We added the p value in the abstract and ‘statistical analysis’ sections. In this review, "p value" refers to the statistical significance of the heterogeneity test, which is typically reported using the Q statistic and associated p-value. 

2) In point 1 of the inclusion criteria, please use the abbreviation of ‘years of potential life lost’ as YPLL since the expanded form has been used earlier in the manuscript. 

Thank you for comment. The sentence was edited with YPLL. Throughout the manuscript, we also used the term SEYLL (standard expected years of life lost) together with YPLL. SEYLL is the other years of life lost method proposed by the GBD study. Because SEYLL is also widely used, we included it in our review. So, we planned to measure both estimated YPLL (by Gardner's method) and SEYLL (from the GBD study). 

3) From point 4 of the inclusion criteria, the authors should consider removing “We will also consider including any” from the rest of the phrase for maintaining the structure of this section. 

Thank you for your comment. The sentence for “We will also consider including any” was removed. 

4) In the ‘discussion’ section, the authors have described the limitations of using observational studies as “Such studies have limitations in drawing precise inferences”. However, this is a vague point and can be replaced by “observational studies can be more prone to bias and confounding”. 

We appreciate your comment and agree with your point. We changed the sentence as suggested.

---

## [Editor Report · Decision Letter 1]

22 Mar 2023

Premature mortality and years of potential life lost from cardiovascular diseases: protocol of a systematic review and meta-analysis

PONE-D-22-21753R1

Dear Dr. Rodzlan Hasani,

We’re pleased to inform you that your manuscript has been judged scientifically suitable for publication and will be formally accepted for publication once it meets all outstanding technical requirements.

Kind regards,

Ahmed Mustafa Rashid

Academic Editor

PLOS ONE
---

## [Editor Report · Acceptance letter]

24 Apr 2023

PONE-D-22-21753R1 

Premature mortality and years of potential life lost from cardiovascular diseases: protocol of a systematic review and meta-analysis 

Dear Dr. Rodzlan Hasani:

I'm pleased to inform you that your manuscript has been deemed suitable for publication in PLOS ONE. Congratulations! Your manuscript is now with our production department. 

Kind regards, 

on behalf of

Dr. Ahmed Mustafa Rashid 

Academic Editor

PLOS ONE